# Perceptions toward Ebola vaccination and correlates of vaccine uptake among high-risk community members in North Kivu, Democratic Republic of the Congo

Shiromi M. Perera[1]*, Stephanie Chow Garbern[2], Eta Ngole Mbong[3], Monica K. Fleming[4], Rigobert Fraterne Muhayangabo[3], Arsene Baleke Ombeni[3], Shibani Kulkarni[4], Dieula Delissaint Tchoualeu[4], Ruth Kallay[4], Elizabeth Song[5], Jasmine Powell[5], Monique Gainey[6], Bailey Glenn[4,7], Ruffin Mitume Mutumwa[3], Stephane Hans Bateyi Mustafa[8], Giulia Earle-Richardson[9], Rena Fukunaga[10], Neetu Abad[4], Gnakub Norbert Soke[11], Dimitri Prybylski[4], David L. Fitter[4], Adam C. Levine[2], Reena H. Doshi[4]

1 International Medical Corps, Washington, District of Columbia, United States of America, 2 Department of Emergency Medicine, Brown University, Providence, Rhode Island, United States of America, 3 International Medical Corps, Goma, Democratic Republic of the Congo, 4 Global Immunization Division, Centers for Disease Control and Prevention, Atlanta, Georgia, United States of America, 5 Brown University, Providence, Rhode Island, United States of America, 6 Rhode Island Hospital, Providence, Rhode Island, United States of America, 7 James A. Ferguson Infectious Disease Program, Centers for Disease Control and Prevention, Atlanta, Georgia, United States of America, 8 Expanded Programme on Immunization, Goma, Democratic Republic of the Congo, 9 National Center for Emerging and Zoonotic Infectious Diseases, Centers for Disease Control and Prevention, Atlanta, Georgia, United States of America, 10 Division of Global HIV and TB, Centers for Disease Control and Prevention, Atlanta, Georgia, United States of America, 11 Division of Global Health Protection, Centers for Disease Control and Prevention, Kinshasa, Democratic Republic of the Congo

probY These authors contributed equally to this work.
* sperera@internationalmedicalcorps.org

## Abstract

The tenth Ebola Virus Disease (EVD) outbreak (2018–2020, North Kivu, Ituri, South Kivu) in the Democratic Republic of the Congo (DRC) was the second-largest EVD outbreak in history. During this outbreak, Ebola vaccination was an integral part of the EVD response. We evaluated community perceptions toward Ebola vaccination and identified correlates of Ebola vaccine uptake among high-risk community members in North Kivu, DRC. In March 2021, a cross-sectional survey among adults was implemented in three health zones. We employed a sampling approach mimicking ring vaccination, targeting EVD survivors, their household members, and their neighbors. Outbreak experiences and perceptions toward the Ebola vaccine were assessed, and modified Poisson regression was used to identify correlates of Ebola vaccine uptake among those offered vaccination. Among the 631 individuals surveyed, most (90.2%) reported a high perceived risk of EVD and 71.6% believed that the vaccine could reduce EVD severity; however, 63.7% believed the vaccine had serious side effects. Among the 474 individuals who had been offered vaccination, 397 (83.8%) received the vaccine, 180 (45.3%) of those vaccinated received the vaccine after two or more offers. Correlates positively associated with vaccine uptake included having heard positive information about the vaccine (RR 1.30, 95% CI 1.06–1.60), the belief that the

**Data Availability Statement:** Due to the sensitive nature of the data used, we request an exemption to making the dataset publicly available. Please use the following Data Availability Statement: The limited, de-identified dataset used to produce this research can currently be requested from the Research, Evidence, and Learning Unit at International Medical Corps by submitting an email to sperera@internationalmedicalcorps.org.

**Funding:** The funding for this work was received through a Cooperative Agreement (NU2GGH002058) from the U.S. Centers for Disease Control and Prevention to International Medical Corps (SMP, SCG, ENM, RFM, ABO, ES, JP, MG, RMM, ACL). U.S. Centers for Disease Control and Prevention authors had a role in the study design, data collection and analysis, decision to publish, and preparation of the manuscript.

**Competing interests:** The authors have declared that no competing interests exist.

vaccine could prevent EVD (RR 1.23, 95% CI 1.09–1.39), and reporting that religion influenced all decisions (RR 1.13, 95% CI 1.02–1.25). Ebola vaccine uptake was high in this population, although mixed attitudes and vaccine delays were common. Communicating positive vaccine information, emphasizing the efficacy of the Ebola vaccine, and engaging religious leaders to promote vaccination may aid in increasing Ebola vaccine uptake during future outbreaks.

## Introduction

Successive Ebola virus disease (EVD) outbreaks in the Democratic Republic of the Congo (DRC) have emphasized the necessity of Ebola vaccines for outbreak response. EVD outbreaks are challenging and require an integrated response that can rapidly identify and isolate suspect cases, trace contacts, organize risk communication and community engagement (RCCE) activities, conduct safe and dignified burials, and administer vaccines [1–3]. Since 2018, the rVSVΔG-ZEBOV-GP (ERVEBO®) vaccine has been used regularly in response to outbreaks caused by Zaire ebolavirus and has shown to be safe and effective [4].

The tenth EVD outbreak in the DRC (2018–2020) was the second largest in history, lasting more than two years and spreading to three provinces (North Kivu, South Kivu, and Ituri), causing over 3,480 cases and 2,200 deaths [5, 6]. At the time of the outbreak, the unlicensed Ebola vaccine was the only vaccine approved for use [7]. Vaccination was offered using a ring vaccination approach, where individuals were eligible for vaccination with investigational doses under an Expanded Access/Compassionate use protocol as recommended by the World Health Organization (WHO)'s Strategic Advisory Group of Experts on Immunization (SAGE) [7–9]. A ring was defined as traceable contacts of an EVD case within a transmission cluster and their contacts, which was meant to create a protective "ring" or cluster of immune individuals around an EVD case to prevent further transmission. More than 300,000 people were vaccinated between August 2018 and June 2020 [10, 11].

Throughout the outbreak, response activities, including vaccination, were complicated by the complex humanitarian crisis in the region (i.e, active conflict, multiple armed groups, and massive population displacement). Coordinated response efforts, led by the DRC government and other humanitarian aid organizations, faced substantial resistance to outbreak control, due to attacks on response workers by armed groups, insecurity, inter-ethnic fighting, sociopolitical unrest, and community mistrust in the government and the response [12, 13]. Rumors and misinformation about EVD and Ebola vaccination spread throughout the community and social media platforms [14, 15]. These challenges contributed to reduced confidence in the response and difficulties with vaccination, including the enumeration and follow-up of contacts, community resistance, and vaccine refusals [1–3].

Vaccine confidence involves trust in vaccine safety and efficacy as well as trust in health systems that deliver the vaccine [16]. Vaccine confidence can be an important driver of vaccine uptake [17]. Factors of vaccine confidence such as trust in those offering the vaccines, and the belief that the vaccine could prevent Ebola transmission during the West African outbreak were related to vaccine uptake [18–21].

Developing a deeper understanding of the public perceptions of the Ebola vaccine, vaccine confidence, and the sociodemographic and behavioral determinants of vaccine uptake can drive interventions aimed at increasing vaccine confidence. Addressing barriers to vaccination will be critical for informing future EVD outbreak response interventions [22, 23]. As such,

this assessment aimed to understand the community members' perceptions towards Ebola vaccination and identify the main correlates of Ebola vaccination uptake during the 2018–2020 EVD outbreak in North Kivu, DRC.

## Materials and methods

### Survey setting and study design

North Kivu is one of 26 provinces located in the Northeastern DRC and has experienced decades of conflict and security issues. The population is known for its strong distrust of both the government and foreigners [12, 24, 25].

In March 2021, we conducted a cross-sectional assessment in North Kivu Province, DRC among adult community members (>18 years) who were likely eligible for the Ebola vaccine as part of the ring vaccination approach during the 10th EVD outbreak. This included, EVD survivors, their household members, and members of survivors' neighboring households. Three health zones (Beni, Butembo, and Mabalako), as shown in S1 Fig, were selected due to their high case counts and persistent community resistance to response activities. We used a modified cluster sampling strategy mimicking ring vaccination to enroll individuals. Sample size methods for assessing a proportion in a two-stage cluster survey (i.e. $n = DEFF*Z^2*p*(1-p)/d^2$) with an estimated design effect (DEFF) of 2.5 were used, assuming an intracluster correlation of 0.167 [26, 27]. To allow for a precision (d) of 7.5% around each of the variables included in the survey regardless of their individual proportions (p, estimated at 0.5), with a confidence interval of 95% (Z = 1.96), a sample size of at least 426 individuals was required, divided up into 39 clusters of approximately 11 people each (neighbors, household contacts, plus the survivor).

The local voluntary community EVD survivors'association provided a list of all EVD survivors, from which 39 persons were randomly selected as a point of reference for the clusters. All adult members of the selected survivors' households were approached for enrollment. As the number of eligible adult household members in each household was not known in advance, ten households surrounding the survivor household were also included in each cluster to ensure a minimum of ten adults per survivor cluster. To avoid bias, all adults in selected households were enrolled, even if that resulted in more than 10 adults per cluster. A spin-the-bottle technique was used to select neighboring households with this process and repeated until at least ten adult participants were enrolled in each survivor cluster.

Participants were eligible if they were at least 18 years of age and had lived in Beni, Butembo, or Mabalako health zones during the outbreak. Individuals who had moved to the area only after the end of the outbreak and those who had lived with a survivor only after their recovery or had never heard of EVD were excluded.

### Data collection

Data collectors (10 in Beni, 10 in Butembo, and 8 in Mabalako), that were not affiliated with the government, and three supervisors per health zone received a four-day training. Data collectors worked in pairs (male and female) so that participants were interviewed by someone of the same sex. Questionnaires were in French, but data collectors were local to the area and were able to translate to Swahili on an ad hoc basis as needed. Translation from French to Swahili was practiced by data collectors during the training. Survey instruments were pilot tested in a community near the training site, which was not part of the sampling frame or included in the analysis. Recruitment of participants began on March 3, 2021 and ended March 15, 2021. All data collectors were required to adhere to prevention measures (i.e., social distancing and

the use of appropriate personal protective equipment) because of the ongoing COVID-19 pandemic.

## Survey

The survey instruments have been described elsewhere [28]. Briefly, the questionnaire included the following topics: respondent demographics, knowledge and perceptions toward EVD and the Ebola vaccine, and attitudes toward general vaccine confidence (i.e., perceptions toward routine immunizations). The questionnaire was translated into French and digitized, using Kobo Toolbox and uploaded to tablets [29].

## Data analysis

Descriptive analyses, using frequencies with percentages, medians with interquartile ranges (IQR), or means with standard deviations (SD) were performed as appropriate. The cluster variable was defined as a survivor, their household members, and the members of neighboring households. A modified Poisson regression model using STATA's 'xtgee' procedure was used to assess potential associations of independent explanatory variables with the primary outcome of vaccine uptake among the respondents that were eligible and offered the Ebola vaccine. Modified Poisson regression has been suggested as a preferable alternative to binomial regression due to easier interpretability of relative risks (versus odds ratios) and due to improved approximation of risk when the outcome is not rare [30, 31] Stata Version 16 (StataCorp, College Station, USA) was used for all analyses.

## Multivariable analysis

Vaccine uptake was measured, using a binary variable indicating whether the respondent received the Ebola vaccine or not. Vaccination status was determined through either verification of the respondent's vaccination certificate or verbal recall. Based on a literature review of existing models such as the 3C, 5C, and health belief models for vaccine hesitancy, explanatory variables were selected for inclusion in the regression model [32] Current understanding of vaccine hesitancy suggests factors such as beliefs regarding vaccine safety/efficacy, perception of risk, desire to protect oneself and ones' community, as well as trust and confidence in the vaccine are key drivers of vaccine hesitancy [32, 33] Variables in the model included sociodemographic variables (sex, age, education level, influence of religion [no influence, some influence, influences all decisions]), perceived risk of contracting Ebola during the outbreak, hearing positive or negative information about the vaccine, vaccine safety, vaccine efficacy, and trust in vaccine source or how it was produced. A composite score for general vaccine confidence was computed, using six items (S1 Table) that have been previously validated in Sierra Leone [34]. Each question had a scale of 0–3 corresponding to low-high vaccine acceptance. The total composite score (range 0–18) was then categorized as low ($<25^{th}$ percentile), medium (25–75$^{th}$ percentile), or high vaccine ($>75^{th}$ percentile) acceptance.

## Ethics statement

The University of Kinshasa School of Public Health Ethics Committee approved the survey (protocol approval #203–2020). Verbal informed consent was obtained and documented electronically because of low literacy rates and the need to limit physical contact during the COVID-19 pandemic. Participation was anonymous, voluntary, and uncompensated.

## Results

### Respondent characteristics

A total of 631 individuals met the inclusion criteria and consented to participate. The median age was 31 years (IQR 22–42; range 18–88) with 423 (67%) females (Table 1). More than half (380; 60.2%) of the respondents had at least some secondary school education. There were 39 EVD survivors, 45 (7.1%) members of the survivors' households, and 547 (86.7%) neighbors of survivors. Further characteristics of the survey respondents are detailed in Table 1.

### Outbreak experiences

Most (514; 81.5%) respondents perceived themselves to be at risk of contracting EVD during the tenth outbreak, with nearly all (590; 93.5%) reporting awareness of someone in their village who had contracted EVD. Approximately half (348; 55.2%) reported direct contact with someone with EVD while they were ill or had attended the funeral of a person diagnosed with EVD. Almost all respondents (603; 95.6%) were aware of the Ebola vaccination program, and 85 (13.5%) had participated in the EVD response.

### Vaccine information

The majority of respondents reported they had heard both positive (502; 79.6%) and negative information (567; 89.9%) communicated about the Ebola vaccine during the outbreak (Table 2). Most respondents had heard that the vaccine was effective in protecting them from EVD (439; 87.5%) and would protect their community (264; 52.6%). The most common negative information heard was that the vaccine would make one sick (358; 63.1%), cause infertility (320; 56.4%), was unsafe (310; 54.7%), had side effects (295; 52.0%) and would lead to death (73; 12.9%). Other negative information respondents heard was that the vaccine is experimental, contaminated, and not accepted by religious leaders. A few respondents reported hearing that healthcare personnel receive a different vaccine than the rest of the population.

### Perceptions toward Ebola virus disease and Ebola vaccination

Nearly all (569; 90.2%) respondents perceived EVD to be a serious and potentially fatal disease (Table 3). Slightly more than half (352; 55.6%) strongly agreed or agreed that vaccination could prevent EVD, whereas a majority (452; 71.6%) strongly agreed or agreed that the vaccine could reduce EVD severity. However, nearly two-thirds (402; 63.7%) believed that the vaccine has severe side effects. Mistrust was relatively common with 223 (35.3%) reporting mistrust of the vaccination team and 245 (38.8%) reporting mistrust of the vaccine source. Many respondents, 272 (43.1%), felt that new vaccines posed more risk and 211 (33.4%) reported mistrust in the government's ability to make decisions about vaccines.

### Ebola vaccine eligibility and uptake

A total of 474 (75.1%) respondents reported they were eligible and were offered the vaccine, and 397 (83.8%) of those eligible accepted the vaccine (Table 4). Of those accepting the vaccine, vaccination status was determined through verification of vaccine certificate for 58 (14.6%) and by verbal recall for 339 (85.4%). Among the 397 vaccinated respondents, 208 (52.4%) received the vaccine upon the first offer. Vaccine delay was common with 180 (45.3% of vaccinated respondents) reporting they received the vaccine only after two or more offers. However, 333 (83.9%) of vaccinated respondents stated they would recommend the vaccine to others. Of those vaccinated, most reported that the benefits of vaccination were explained to them (351; 88.4%), as well as potential side effects (354; 89.2%), at the time of vaccination.

**Table 1. Sociodemographic characteristics of the surveyed community members, North Kivu, Democratic Republic of the Congo, 2021.**

| Characteristic | (N = 631) n (%) |
|---|---|
| **Age** (years), median [IQR] | 31 [22–42] |
| **Sex** | |
| Male | 208 (33.0) |
| Female | 423 (67.0) |
| **Health Zone** | |
| Beni | 239 (37.9) |
| Butembo | 250 (39.6) |
| Mabalako | 142 (22.5) |
| **Respondent Type** | |
| EVD Survivor | 39 (6.2) |
| Member of EVD Survivor Household | 45 (7.1) |
| Neighbor of EVD Survivor | 547 (86.7) |
| **Highest Education Level** | |
| None | 72 (11.4) |
| Primary school | 175 (27.7) |
| Secondary school | 324 (51.3) |
| University or Higher Institute | 56 (8.9) |
| Don't know / Declined | 4 (0.6) |
| **Religion** | |
| Catholic | 352 (55.8) |
| Protestant/Evangelical/Pentecostal/Revival | 250 (39.6) |
| Muslim | 17 (2.7) |
| Animist | 4 (0.6) |
| Atheist | 4 (0.6) |
| Other | 1 (0.2) |
| Declined | 3 (0.5) |
| **Influence of Faith on Decisions Including Health** | |
| No influence | 213 (33.8) |
| Influences some decisions | 205 (32.5) |
| Influences all decisions | 207 (32.8) |
| Declined to Respond | 6 (1.0) |
| **Primary Occupation** | |
| Farmer | 181 (28.7) |
| Unemployed | 113 (17.9) |
| Homemaker | 93 (14.7) |
| Student | 68 (10.8) |
| Trader / Businessperson | 69 (10.9) |
| Healthcare Worker | 24 (3.8) |
| Work from home | 21 (3.3) |
| Teacher | 9 (1.4) |
| Other* | 53 (8.4) |

*Other occupations (each listed occupation with less than five responses): Fisherman, Traditional healer, Seamstress, Carpenter, Driver, Electrician, Gardener, Engineer, Plumber, Mason, Shoemaker, Military personnel.

**Table 2. Positive and negative information heard about the Ebola vaccine, North Kivu, Democratic Republic of the Congo, 2021.**

|  | N (%) |
|---|---|
| **Positive Information heard about the Ebola vaccine** | **N = 631** |
| Yes | 502 (79.6) |
| No | 102 (16.2) |
| Declined to respond | 27 (4.3) |
| *If you heard positive information, what information did you hear?** | **N = 502** |
| The vaccine is effective in protecting you from EVD | 439 (87.5) |
| The vaccine will protect my community | 264 (52.6) |
| The vaccine is good for you | 211 (42.0) |
| The vaccine is safe | 189 (37.7) |
| Other | 12 (2.4) |
| **Negative Information heard about the Ebola vaccine** | **N = 631** |
| Yes | 567 (89.9) |
| No | 50 (7.9) |
| Declined to respond | 14 (2.2) |
| *If you heard negative information, what information did you hear?** | **N = 567** |
| The vaccine makes you sick | 358 (63.1) |
| The vaccine causes infertility | 320 (56.4) |
| The vaccine is not safe | 310 (54.7) |
| The vaccine has side effects | 295 (52.0) |
| The vaccine gives you Ebola | 255 (45.0) |
| The vaccine is harmful to babies in pregnant women | 147 (25.9) |
| The vaccine will kill us | 73 (12.9) |
| Other | 53 (9.3) |

* Multiple selections were allowed; therefore, total proportions do not sum to 100%.

Among the 77 respondents who were eligible but declined vaccination, the most common reasons for declining the vaccine included: the belief that the vaccine would make them sick (13; 16.9%), did not believe the vaccine was safe (11; 14.3%), the belief that Ebola was not real (10; 13%), did not feel at risk of EVD (9; 11.7%), the belief the vaccine would give them EVD (9; 11.7%), and not having enough information about the vaccine (8; 10.4%) (Table 5). However, 14 (18.2%) reported they would take the vaccine now if offered, although most (59; 76.6%) reported they still would not take the vaccine. Most unvaccinated respondents (71; 92.2%) indicated that a social or religious group did not influence their decision to take the vaccine.

## General vaccine confidence

Respondents had overall high general vaccine confidence (i.e., perceptions toward routine immunizations) with 460 (72.9%) very much or somewhat agreeing that vaccines were good, and 488 (77.3%) very much or somewhat agreeing that vaccines protect against diseases (S1 Table). The median [IQR] of the general vaccine confidence composite score was 12 [9–15] (out of a maximum of 18) with 225 (47.7%) respondents categorized as having high vaccine acceptance, 191 (40.5%) with moderate vaccine acceptance, and 56 (11.9%) with low vaccine acceptance. There was no significant difference found between the mean general vaccine confidence score between those who received the vaccine and those who declined (11.9 vs 11.4, respectively; p = 0.34).

**Table 3. Beliefs and attitudes\* toward Ebola, the Ebola vaccine, and vaccines in general, North Kivu, Democratic Republic of the Congo, 2021.**

| Question | Strongly Agree | Agree | Neutral | Disagree | Strongly Disagree | Unsure/ Declined |
|---|---|---|---|---|---|---|
| **n (%) (N = 631)** | | | | | | |
| **BELIEFS** | | | | | | |
| EVD is a serious and potentially fatal disease | 431 (68.3) | 138 (21.9) | 17 (2.7) | 15 (2.4) | 13 (2.1) | 17 (2.7) |
| The vaccine is needed to prevent disease spread during an outbreak | 285 (45.2) | 214 (33.9) | 44 (7.0) | 34 (5.4) | 26 (4.1) | 28 (4.4) |
| Vaccination prevents Ebola Virus Disease | 164 (23.0) | 188 (29.8) | 72 (11.4) | 126 (20.0) | 44 (7.0) | 37 (5.9) |
| The vaccine reduces disease severity | 232 (26.8) | 220 (34.9) | 55 (8.7) | 49 (7.8) | 33 (5.2) | 42 (6.7) |
| The vaccine has severe side effects | 173 (27.4) | 229 (36.3) | 70 (11.1) | 70 (11.1) | 38 (6.0) | 51 (8.1) |
| I think I am now at risk of contracting Ebola | 56 (8.9) | 137 (21.7) | 116 (18.4) | 172 (27.3) | 78 (12.4) | 72 (11.4) |
| **ATTITUDES** | | | | | | |
| **Ebola Vaccine** | | | | | | |
| I wanted to be vaccinated when the vaccine was available in my community | 150 (23.8) | 214 (33.9) | 55 (8.7) | 108 (17.1) | 85 (13.5) | 19 (3.0) |
| Getting vaccinated makes me feel I don't need to take other precautions to protect myself against Ebola | 33 (5.2) | 46 (7.3) | 0 (54) | 240 (38.0) | 226 (35.8) | 32 (5.1) |
| Many people were vaccinated in my community | 239 (37.9) | 246 (39.0) | 33 (5.2) | 38 (6.0) | 25 (4.0) | 50 (7.9) |
| I did not trust the vaccination team | 90 (14.3) | 133 (21.1) | 98 (15.5) | 185 (29.3) | 92 (14.6) | 33 (5.2) |
| I did not trust the vaccine source or how the vaccine was given | 103 (16.3) | 142 (22.5) | 115 (18.2) | 148 (23.5) | 79 (12.5) | 44 (7.0) |
| **Vaccines in General** | | | | | | |
| Insecurity prevents me from accessing vaccines or other health services | 29 (4.6) | 66 (10.5) | 77 (12.2) | 412 (65.3) | 0 (0) | 47 (7.5) |
| I do not trust the government to make decisions about vaccines | 94 (14.9) | 117 (18.5) | 121 (19.2) | 148 (23.5) | 101 (16.0) | 50 (7.9) |
| New vaccines pose more risk | 134 (21.2) | 138 (21.9) | 110 (17.4) | 76 (12.0) | 60 (9.5) | 113 (17.9) |

\*Based on Likert scale questions.

## Correlates of vaccine uptake

Survivor cluster information was not available for two respondents, leaving 472 respondents for multivariable analysis. Correlates associated with vaccine uptake included having heard positive information communicated about the Ebola vaccine [adjusted risk ratio (aRR) 1.30, 95% CI 1.06–1.60], belief that the vaccine could prevent EVD (aRR 1.23, 95% CI 1.09–1.39), and reporting that religion influences all of one's decisions (versus none; aRR 1.13, 95% CI 1.02–1.25). Demographic factors including sex, age, educational level, general vaccine acceptance, and having heard negative information about the vaccine were not associated with vaccine uptake (Table 6).

## Discussion

Our survey revealed high uptake of the Ebola vaccine among adult community members in three health zones heavily affected during the 10th EVD outbreak in North Kivu, DRC. Our

**Table 4. Ebola vaccine eligibility, uptake, and the number of offers prior to vaccine receipt, North Kivu, Democratic Republic of the Congo, 2021.**

| Ebola Vaccine Eligibility and Vaccination Status | n(%) |
|---|---|
| **Eligibility and Vaccine Offers** | **N = 631** |
| Eligible and offered opportunity to receive vaccine | 474 (75.1) |
| Ineligible or not offered vaccine* | 157 (24.9) |
| **Vaccine Uptake†** | **N = 474** |
| Received vaccine | 397 (83.8) |
| Declined vaccine | 77 (16.2) |
| **Number of Vaccine Offers Prior to Vaccine Receipt** | **N = 397** |
| Vaccinated at first offer | 208 (52.4) |
| Vaccinated at second offer | 71 (17.9) |
| Vaccinated at third offer | 47 (11.8) |
| Vaccinated at fourth offer or later | 62 (15.6) |
| Do not recall | 9 (2.3) |

* Not offered or were informed they were ineligible, per patients' verbal recall. We did not solicit information about the specific reasons for patients being informed of their ineligibility.

† Among those eligible and offered vaccination only.

findings are consistent with other studies in this region, including our recent work demonstrating very high vaccine uptake among healthcare workers [28, 35–37]. While many survey respondents believed the vaccine to be effective and important to prevent the spread of EVD in their community, mixed attitudes toward the vaccine among both vaccinated and

**Table 5. Reasons for declining vaccination among eligible but unvaccinated community members, North Kivu, Democratic Republic of the Congo, 2021.**

| | n(%) N = 77 |
|---|---|
| **Reasons for Declining Vaccination*** | |
| I thought the vaccine was going to make me sick | 13 (16.9) |
| I did not think the vaccine was safe | 11 (14.3) |
| Ebola is not real, so the vaccine is not needed | 10 (13) |
| I did not feel at risk for Ebola | 9 (11.7) |
| I thought the vaccine was going to give me Ebola | 9 (11.7) |
| I did not have enough information about the vaccine | 8 (10.4) |
| I was pregnant or breastfeeding at the time | 7 (9.1) |
| The vaccine was too new (experimental) | 6 (7.8) |
| I didn't think the vaccine was effective at preventing EVD | 5 (6.5) |
| The vaccination site was too far away | 3 (3.9) |
| I did not want to identify myself as eligible to be vaccinated | 3 (3.9) |
| I did not want to sign a form | 2 (2.6) |
| I did not trust the government | 2 (2.6) |
| The times and days when vaccination was offered were not possible for me | 3 (3.9) |
| Vaccination process took too long | 1 (1.3) |
| There were too many changes to the vaccination program/protocol (dose, eligibility changes, pregnant/women, age, etc.) | 1 (1.3) |
| I did not trust the local team that was offering the vaccine | 1 (1.3) |

* Not mutually exclusive; multiple selections were allowed.

**Table 6. Correlates of Ebola vaccine uptake, using modified Poisson regression among community members eligible and offered vaccination during the tenth EVD outbreak, North Kivu, Democratic Republic of the Congo, 2021.**

| | Received Vaccine | Declined Vaccine | RR (95% CI) | aRR† (95% CI) |
|---|---|---|---|---|
| | n(%) N = 395 | n(%) N = 77 | | |
| **Sex** | | | | |
| Male | 129 (32.7) | 28 (36.4) | - | - |
| Female | 266 (67.3) | 49 (63.6) | 1.04 (0.96–1.12) | 1.06 (0.99–1.14) |
| **Age (years), median [IQR]** | 31 [22–41] | 26 [21–38] | 1.00 (1.00–1.01) | 1.00 (1.00–1.01) |
| **Highest Education Attained** | | | | |
| None | 41 (10.4) | 8 (10.4) | - | - |
| Primary | 101 (25.6) | 22 (28.6) | 0.96 (0.82–1.13) | 0.94 (0.79–1.13) |
| Secondary | 206 (52.2) | 55 (57.1) | 0.99 (0.87–1.13) | 1.02 (0.88–1.17) |
| University or Higher | 44 (11.1) | 3 (3.9) | 1.10 (0.96–1.26) | 1.06 (0.89–1.26) |
| Missing / Declined | 3 (0.8) | 0 (0) | - | - |
| **Religion Influence** | | | | |
| No influence | 125 (31.7) | 32 (41.6) | - | - |
| Influences some decisions | 121 (30.6) | 29 (37.7) | 1.05 (0.93–1.18) | 1.04 (0.93–1.16) |
| Influences all decisions | 147 (37.2) | 15 (19.5) | 1.13 (1.02–1.24) | 1.13 (1.02–1.25)* |
| Missing / Declined | 2 (0.5) | 1 (1.3) | | |
| **Perceived Risk of EVD during outbreak** | | | | |
| No / Do not recall | 41 (10.4) | 15 (19.5) | - | - |
| Yes | 354 (89.6) | 62 (80.5) | 1.22 (0.98–1.51) | 1.10 (0.91–1.34) |
| **Heard positive information about the vaccine** | | | | |
| No | 50 (12.7) | 25 (32.5) | - | - |
| Yes | 345 (87.3) | 52 (67.5) | 1.38 (1.12–1.69) | 1.30 (1.06–1.60)* |
| **Heard negative information about the vaccine** | | | | |
| No | 43 (10.9) | 6 (7.8) | - | - |
| Yes | 352 (89.1) | 71 (92.2) | 0.93 (0.81–1.08) | 0.94 (0.81–1.11) |
| **EVD can be prevented with vaccine** | | | | |
| No | 123 (31.1) | 53 (68.8) | - | - |
| Yes | 272 (68.8) | 24 (31.2) | 1.29 (1.15–1.46) | 1.23 (1.09–1.39)* |
| **Ebola vaccine has severe side effects** | | | | |
| No / Unsure | 143 (36.2) | 27 (35.1) | - | - |
| Yes | 252 (63.8) | 50 (64.9) | 0.96 (0.87–1.07) | 0.95 (0.85–1.06) |
| **Mistrust of vaccine source or how it was given** | | | | |
| No | 263 (66.6) | 44 (57.1) | - | - |
| Yes | 132 (33.4) | 33 (42.9) | 0.92 (0.86–1.00) | 0.96 (0.90–1.04) |
| **General Vaccine Confidence** | | | | |
| Low | 49 (12.4) | 7 (9.1) | - | - |
| Medium | 150 (38.0) | 41 (52.3) | 0.94 (0.84–1.05) | 0.92 (0.82–1.04) |
| High | 196 (49.6) | 29 (37.7) | 1.03 (0.92–1.15) | 0.94 (0.83–1.07) |

*Only respondents who were eligible and offered vaccination were included (n = 415).

†aRR = adjusted risk ratio

unvaccinated respondents were common. Notably, while most felt the vaccine was needed to prevent disease spread during an outbreak, 20% disagreed that the vaccine prevents EVD. This might be explained by general understanding that no vaccine is 100% effective or perception that the vaccine is still experimental. It is also possible that this discrepancy is attributed to knowledge of breakthrough infections, which did occur, especially among contacts who were

likely vaccinated during their incubation period. Nearly one-third of respondents said they did not want to receive the vaccine when it was first available, suggesting early low vaccine confidence and highlighting the importance of reoffering Ebola vaccines, and continuous RCCE strategies that build confidence in vaccine safety and efficacy. More nuanced understanding of additional motivations for vaccination, such as social desirability or fear of response teams, would be best addressed using qualitative research methods, which are planned.

This population indicated a high perceived risk of contracting EVD, with most respondents indicating that they knew someone with EVD in their village, and approximately half had direct contact with an EVD case. This was expected, given the survey took place in areas with large numbers of EVD cases and we attempted to target contacts and contacts of contacts who were likely to be offered Ebola vaccination. Unlike other studies that mostly surveyed affected communities during the early stages of the outbreak, our survey assessed perceptions toward the Ebola vaccination at the end of the outbreak in areas that were frequent Ebola epicenters. As a result, our respondents, who were at high-risk for EVD infection, had prior experience with Ebola and were heavily targeted during the course of the outbreak by Ebola vaccination campaigns. Perceived risk is closely associated with willingness to receive various vaccines, including the Ebola vaccine. This is a key component of various models used to explain vaccine-related behavior such as the Health Belief Model and 5C model of vaccine hesitancy [13, 19, 33, 38, 39]. Perceived risk of EVD was not significantly associated with vaccine uptake in the multivariable analysis. This may be explained by the fact that the entire survey sample consisted of persons who were more likely to be part of a"ring,"making it difficult to ascertain differences.

Respondents who heard positive instead of negative information about the Ebola vaccine were more likely to accept the vaccine when offered. This finding aligns with the currently recommended vaccine communication strategies [40]. Positive vaccine information and recommendations from authorities have been shown to increase vaccine confidence and acceptance, while negative messaging, including belief in misinformation and rumors, have been associated with decreased willingness [1, 13, 41]. Rumors and misinformation during the outbreak were widespread; social media platforms facilitated the rapid spread [42]. Rumors such as the vaccine could lead to infertility, cause EVD, and that the vaccine was introduced intentionally to sterilize and depopulate the region were common [1]. Therefore, positive and transparent communication about the benefits of vaccination and dispelling negative rumors and harmful misinformation through multiple communication channels and approaches are crucial.

There were mixed attitudes toward the vaccine, with two-thirds indicating they were concerned about potential severe side effects. This is not surprising, given the vaccine frequently causes mild-to-moderate side effects, such as fever, arthralgia, myalgia, fatigue, and headache [43]. More than a third of respondents reported mistrust of the vaccine source. The vaccine had not been used extensively in DRC and it was initially unlicensed; hence, investigational doses were being used under a compassionate use, expanded access protocol [44, 45]. Vaccination required informed consent and active safety monitoring for adverse events, which contributed to concerns about the experimental nature of the vaccine, despite its safety and effectiveness shown in clinical trials during the 2014–2016 West African outbreak [2, 4, 46, 47]. Moreover, the eligibility criteria were revised to include pregnant women (after the first trimester), lactating women, and children 6 months and older [48]. Concerns about the low vaccine supply resulted in the use of fractional doses. Additionally, a second Ebola vaccine, a two-dose regime, was offered as part of a clinical trial in an unaffected area near Goma, North Kivu [49–51]. All these changes may have resulted in confusion and distrust in the Ebola vaccination program. Respondents also indicated mistrust of the government and their handling of the 2018–2020 EVD response, as well as reports of security issues affecting vaccine access.

Several respondents questioned whether Ebola was real, which is consistent with other work in DRC, indicating community perceptions that Ebola might have been fabricated for financial gains or to destabilize the region [1, 13, 41]. North Kivu has a complex sociopolitical environment and ongoing violence; security issues led to tension and a decline in trust toward the government [13]. Politicization of the EVD response, deliberate circulation of misinformation for political gain, and suspicion toward response workers including the vaccination teams, may have contributed to concerns about the vaccine [13, 35, 52]. These findings are consistent with EVD studies in West Africa and the DRC, showing that community resistance and the lack of trust in the government impact compliance with EVD control measures and policies, such as vaccination [13, 53].

Vaccine uptake is influenced by a diverse set of individual-level and community-level factors and vaccine-specific issues [54–56]. In our multivariable analysis, we found that religious influence on decision-making and having heard positive information about the vaccine were associated with increased vaccine uptake. In the DRC, religious leaders are trusted and respected figures who may influence community members' attitudes and beliefs toward vaccination [1]. Prior research across 13 countries demonstrated the influential role of religious leaders in influencing vaccine acceptance. A 2019 household survey conducted in Sierra Leone found that the promotion of vaccination by religious leaders was associated with an increased likelihood of Ebola vaccine uptake [27, 57]. Incorporation of religious leaders in community sensitization campaigns can be used to build vaccine confidence and convey positive information about the vaccine during an outbreak.

The belief that the Ebola vaccine was effective was also associated with increased vaccine uptake. Belief in vaccine efficacy has also been shown to increase willingness to receive the Ebola vaccine in studies from both North Kivu as well as during the West African EVD outbreak [13, 58]. Interestingly, vaccine safety was not found to be associated with vaccine uptake, despite a large proportion of respondents reporting concerns that the vaccine had severe side effects. However, belief in the vaccine's efficacy and fear of EVD may have outweighed the fear of vaccine safety.

Nearly half of the respondents reported only receiving the vaccine after two or more offers. This finding highlights the importance of repeated efforts to engage the "moveable middle,' those individuals who have concerns regarding vaccines, but may be willing to change their decisions with additional information or influence from other sources [59]. EVD outbreaks are increasing in frequency and vaccination has become an integral part of the response; therefore, timely uptake, especially among the contacts of contacts can be used to break chains of transmission [46, 60]. During future EVD outbreaks, coordinated efforts to "close the ring" or vaccinate all contacts (and contacts of contacts) by focusing on understanding and addressing the concerns of individuals who intend to delay vaccination will be crucial to halt the spread of the disease. Continuing RCCE efforts with targeted messages that build Ebola vaccine confidence and address the remaining concerns of those who delay vaccination or refuse are important for designing interventions in future EVD and other infectious disease outbreaks.

DRC is prone to outbreaks of other multiple vaccine-preventable diseases, such as measles, polio, cholera, and meningitis [61]. General vaccine confidence was not associated with Ebola vaccine uptake in our survey population, but we did find that their overall vaccine confidence was high. North Kivu routinely outperforms other provinces in routine immunization indicators, which may be explained by the continuous presence of aid organizations [54, 62]. A cross-sectional community survey in DRC suggested that respondents were more likely to accept routine vaccinations (90%) compared to the outbreak (i.e., cholera, Ebola, COVID-19) vaccinations (57%); this may be due to new vaccines being perceived as carrying more risk than routine vaccines or lower perceived disease susceptibility [61]. Lastly, we found that none of the sociodemographic predictors included were associated with vaccine uptake. While gender, age,

education level, and socioeconomic status have been associated with vaccine acceptance in other studies, demographic factors are often highly context-dependent and insufficient to independently explain outcomes of vaccine confidence or acceptance uniformly [56, 57].

Our findings are subject to a number of limitations. Given the ring strategy in a large urban environment, the traditional household or coverage survey methodology would have been unlikely to capture those who were eligible for the vaccine. Thus, we sampled among persons who were likely offered part of the "ring" and survey results are not generalizable to the broader community. Our survey was designed to capture perceptions and attitudes toward Ebola vaccine in a population that was present for the 10th Ebola outbreak. This survey was delayed due to the COVID-19 pandemic and then two subsequent EVD outbreaks (11th and 12th), albeit small, occurred in DRC. Living in an area with multiple EVD outbreaks likely influenced the perceptions and attitudes about disease severity and Ebola vaccination. Additionally, we only targeted participants in three health zones in North Kivu, although the outbreak expanded across other health zones and provinces, including Ituri and South Kivu. There is a possibility of misclassification due to recall inaccuracies and some questions and responses may have been misinterpreted or mistranslated by the interviewers even though the survey tool was translated, piloted, and adapted to the country context. We expect the misclassification to be non-differential and more likely to bias the results toward the null. Despite these limitations, this survey was unique and extensive, exploring the outbreak experiences, perceptions, atttidues and beliefs toward the Ebola vaccine and general vaccine confidence on a vulnerable population at a time of active conflict in the region and during a time with COVID-19 restrictions [2, 13, 61].

## Conclusions

Ebola vaccine uptake was high in this population of high-risk individuals in North Kivu, although mixed attitudes and vaccine delays were common. We identified context-specific correlates of vaccine uptake, including individual, community, and vaccine-specific issues. Interventions focusing on communicating positive vaccine information, especially emphasizing the efficacy of the Ebola vaccine in addition to its safety, and engaging religious leaders to promote vaccination, may aid in increasing Ebola vaccine uptake when employing ring vaccination strategies during future EVD and other infectious disease outbreaks.

## Supporting information

**S1 Fig. Map of health zones (Beni, Butembo, Mabalako) surveyed in North Kivu, Democratic Republic of the Congo, March 2021. Source:** https://data.humdata.org/dataset/rdc-statistiques-des-populations.
(TIFF)

**S1 Table. Perceptions towards routine immunizations among community members, North Kivu, Democratic Republic of the Congo, 2021.**
(DOCX)

**S1 File. Inclusivity in global research questionnaire.**
(DOCX)

## Acknowledgments

The authors thank all study participants and the International Medical Corps staff who were instrumental in collecting the data used in this study. The authors also thank the DRC's

Ministry of Health and the Expanded Programme on Immunization for facilitation of this work. The findings and conclusions in this paper are those of the authors and do not necessarily represent the official position of the U.S. Centers for Disease Control and Prevention or International Medical Corps or any institutions that the authors are affiliated with.

## Author Contributions

**Conceptualization:** Shiromi M. Perera, Stephanie Chow Garbern, Eta Ngole Mbong, Monica K. Fleming, Shibani Kulkarni, Dieula Delissaint Tchoualeu, Giulia Earle-Richardson, Neetu Abad, Dimitri Prybylski, David L. Fitter, Adam C. Levine, Reena H. Doshi.

**Data curation:** Shiromi M. Perera, Stephanie Chow Garbern, Eta Ngole Mbong, Arsene Baleke Ombeni, Elizabeth Song, Jasmine Powell, Monique Gainey, Bailey Glenn.

**Formal analysis:** Shiromi M. Perera, Stephanie Chow Garbern, Shibani Kulkarni, Elizabeth Song, Jasmine Powell, Monique Gainey, Bailey Glenn.

**Investigation:** Eta Ngole Mbong, Rigobert Fraterne Muhayangabo, Arsene Baleke Ombeni, Ruffin Mitume Mutumwa, Stephane Hans Bateyi Mustafa.

**Methodology:** Shiromi M. Perera, Stephanie Chow Garbern, Shibani Kulkarni, Reena H. Doshi.

**Supervision:** Eta Ngole Mbong, Rigobert Fraterne Muhayangabo, Adam C. Levine, Reena H. Doshi.

**Visualization:** Bailey Glenn.

**Writing – original draft:** Shiromi M. Perera, Stephanie Chow Garbern, Eta Ngole Mbong, Shibani Kulkarni, Reena H. Doshi.

**Writing – review & editing:** Monica K. Fleming, Shibani Kulkarni, Dieula Delissaint Tchoualeu, Ruth Kallay, Giulia Earle-Richardson, Rena Fukunaga, Neetu Abad, Gnakub Norbert Soke, Dimitri Prybylski, David L. Fitter, Adam C. Levine.

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
