## [Decision Letter · Decision Letter 0]

25 Aug 2023

PGPH-D-23-01349

Perceptions toward Ebola vaccination and correlates of vaccine uptake among high-risk community members in North Kivu, Democratic Republic of the Congo

Dear Dr. Perera,

Thank you for submitting your manuscript to PLOS Global Public Health. After careful consideration, we feel that it has merit but does not fully meet PLOS Global Public Health’s publication criteria as it currently stands. Therefore, we invite you to submit a revised version of the manuscript that addresses the points raised during the review process.

We look forward to receiving your revised manuscript.

Kind regards,

Vinay Nair Kampalath, MD, DTMH

Guest Editor

Journal Requirements:

2. Please provide separate figure files in .tif or .eps format.

Additional Editor Comments (if provided):

Thank you for this important contribution. Kindly address the comments from both reviewers (but particularly Reviewer 1 regarding their comments on results/discussion).

Reviewers' comments:

Reviewer's Responses to Questions

**Comments to the Author**

1. Does this manuscript meet PLOS Global Public Health’s publication criteria? Is the manuscript technically sound, and do the data support the conclusions? The manuscript must describe methodologically and ethically rigorous research with conclusions that are appropriately drawn based on the data presented.

Reviewer #1: Yes

Reviewer #2: Yes

2. Has the statistical analysis been performed appropriately and rigorously?

Reviewer #1: Yes

Reviewer #2: Yes

3. Have the authors made all data underlying the findings in their manuscript fully available (please refer to the Data Availability Statement at the start of the manuscript PDF file)?

Reviewer #1: Yes

Reviewer #2: Yes

4. Is the manuscript presented in an intelligible fashion and written in standard English?

Reviewer #1: Yes

Reviewer #2: Yes

5. Review Comments to the Author

Reviewer #1: Using a cross-sectional survey, the authors of this piece assess community perceptions towards Ebola vaccination among high-risk individuals who were likely offered an Ebola vaccine in North Kivu, Democratic Republic of the Congo (DRC), during the DRC's 10th Ebola outbreak. They use the survey results to run a multivariate regression that measures associations between vaccination uptake and several covariates. This study is useful in understanding vaccination patterns in this community, and may have useful implications for future vaccination efforts. I have several comments that I would like the authors to address:

Major

Lines 71-72: The authors say here that there is a strong distrust of foreigners and government in this community. How did the authors deal with this? Did they use local colleagues who were non-government-affiliated to do the surveys? This would be good to specify, as it otherwise leaves room for substantial bias in the results.

Line 75: how was “high risk for contracting EVD” defined?

Table 2: There were a sizable number of “Other” responses for “If you heard negative information, what information did you hear?”. It would be interesting to delve into what some of those were.

Table 3: It’s an interesting discrepancy that the strong majority of people believe that the vaccine is needed to prevent disease spread during an outbreak, but that 20% of people disagree that vaccination prevents EVD. It would be interesting to discuss that further, space permitting. Similarly, the result that 30.6% reported not wanting to be vaccinated when the vaccine was available conflicts with the percentage that did get vaccinated, and this would likewise be interesting to comment on.

Correlates of vaccine uptake/Table 6: Did authors test for collinearity in their regression? There are several covariates that I would expect to be highly correlated (e.g., heard negative information about the vaccine and ebola vaccine has severe side effects).

Discussion: In the limitations section, it would be important for the authors to comment on how their results may be affected by the fact that the participants, at the time of study, had lived through an Ebola outbreak. When the outbreak was beginning (and thus when they were offered the vaccinations), their perceptions around vaccinations and disease severity may have been quite different. This could have important implications for new outbreaks when vaccinations are being offered.

Lines 286-287: The authors comment that a two-dose vaccine regime was offered in areas near the active outbreak. Was North Kivu included in this? If so, are the study results taking into account differences between individuals who got one vaccination versus both?

In the Discussion, the authors mention and cite several sources that appear to have done similar work in North Kivu. How does this study differentiate itself?

Minor

Line 38: Remove extraneous )

Lines 56-58: Is there a reference for this?

Throughout: make sure Ebola versus EVD use is consistent.

I suggest reorganizing Table 2 so that the answers for “If you heard positive/negative information, what information did you hear?” are in order based on response percentage

Line 249: Please define RCCE at first use

Line 293-294: What is the timeframe for this?

How does this study differentiate between prior studies in North Kivu that authors mention?

Table S2: Caption says “eligible but unvaccinated,” but it appears as though all participants are included here

Table 3 and S2: I suggest switching out the rows currently included in Table 3 for Attitudes on Vaccines in General with some of those included in Table S2. The responses for "How much do you think that vaccines are good?,” “How much do you think that vaccines are safe?,” and "How much do you think that vaccines protect against diseases?” from Table S2 are particularly interesting, and including them in Table 3 instead of the current rows would allow for comparisons between results for Ebola vaccination and general vaccination.

Reviewer #2: The article has a clear description of the survey setting, study design, data collection, and analysis methods. There are a few suggestions and comments that could enhance the clarity and completeness of the article in general.

One is to provide more information about why the specific health zones were chosen based on their high case counts, so it would be easier to understand the rationale behind the selection.

Other recommendation is to explain more about the "ring vaccination" concept in a bit more detail, as it might not be familiar to all readers.

When describing the sample size calculation, what was the rationale behind the chosen values for parameters like the design effect (DEFF) and the intracluster correlation.

Do the 39 Ebola survivors that were randomly selected served as focal points for recruitment or as a point of reference for the clusters?

Detail the process of translation from French to Swahili by data collectors. This process could introduce translation bias or issues, and it's important to discuss how this was mitigated.

In the data analysis section, consider expanding on the choice of the modified Poisson regression model using 'xtgee' in STATA for analyzing associations. Explain why this method is suitable for the data and research questions.

Consider providing a bit more context on the rationale behind choosing specific explanatory variables for inclusion in the regression model. This would help readers understand the factors that were deemed important in relation to vaccine uptake.

6. PLOS authors have the option to publish the peer review history of their article (what does this mean?). If published, this will include your full peer review and any attached files.

**Do you want your identity to be public for this peer review?** For information about this choice, including consent withdrawal, please see our Privacy Policy.

Reviewer #1: No

Reviewer #2: **Yes: **Natalia Hernandez Morfin

---

## [Decision Letter · Decision Letter 1]

29 Nov 2023

Perceptions toward Ebola vaccination and correlates of vaccine uptake among high-risk community members in North Kivu, Democratic Republic of the Congo

PGPH-D-23-01349R1

Dear Ms. Perera,

We are pleased to inform you that your manuscript 'Perceptions toward Ebola vaccination and correlates of vaccine uptake among high-risk community members in North Kivu, Democratic Republic of the Congo' has been provisionally accepted for publication in PLOS Global Public Health.

Best regards,

Vinay Nair Kampalath, MD, DTMH

Guest Editor

Reviewer Comments (if any, and for reference):

Reviewer's Responses to Questions

**Comments to the Author**

1. If the authors have adequately addressed your comments raised in a previous round of review and you feel that this manuscript is now acceptable for publication, you may indicate that here to bypass the “Comments to the Author” section, enter your conflict of interest statement in the “Confidential to Editor” section, and submit your "Accept" recommendation.

Reviewer #1: All comments have been addressed

Reviewer #2: All comments have been addressed

2. Does this manuscript meet PLOS Global Public Health’s publication criteria? Is the manuscript technically sound, and do the data support the conclusions? The manuscript must describe methodologically and ethically rigorous research with conclusions that are appropriately drawn based on the data presented.

Reviewer #1: Yes

Reviewer #2: Yes

3. Has the statistical analysis been performed appropriately and rigorously?

Reviewer #1: Yes

Reviewer #2: Yes

4. Have the authors made all data underlying the findings in their manuscript fully available (please refer to the Data Availability Statement at the start of the manuscript PDF file)?

Reviewer #1: Yes

Reviewer #2: Yes

5. Is the manuscript presented in an intelligible fashion and written in standard English?

Reviewer #1: Yes

Reviewer #2: Yes

6. Review Comments to the Author

Reviewer #1: The authors have responded to each of my comments thoroughly and appropriately. Well done! The only remaining suggestion I have, which is very minor, is to mention what the general vaccine confidence composite score is out of in lines 239 - 240, as this would help put their findings into perspective. However, their piece is ready to be accepted.

Reviewer #2: 1) The authors have adequately addressed the reviewers' previous comments. The changes made have improved the manuscript, such as providing more details on the sampling methods, data collection, and analysis approach. The revisions have clarified aspects of the study design, results, and limitations. Overall, the authors were responsive to the feedback provided during the initial review.

2) Yes, this manuscript meets PLOS Global Public Health's publication criteria. The cross-sectional survey employs sound epidemiologic methods to assess community perceptions and uptake of Ebola vaccination in the DRC. The data appear to support the main findings and conclusions related to positive vaccine attitudes but delays in uptake. The study also identified correlates associated with increased vaccine acceptance.

3) The statistical analysis seems appropriate for the study design and objectives. The authors use descriptive statistics to summarize responses and modified Poisson regression to assess correlates of vaccine uptake. They mention checking for collinearity between explanatory variables and provide information on model diagnostics. The analytical methods are suitable for the collected data.

4) The authors state that they plan to publish the de-identified dataset pending approval from the DRC Ministry of Health. This seems reasonable given the sensitive nature of the survey data. The authors agree to provide a link to the data when available. This satisfies the journal's data availability requirements.

5) Overall, the manuscript is well-written in clear English. The background provides context on Ebola outbreaks and response in DRC. The methods and results are presented in a logical manner. The tables and figure effectively summarize key data. The discussion interprets the findings, compares to prior literature, acknowledges limitations, and draws reasonable conclusions. Only minor edits would be needed to further improve clarity.

7. PLOS authors have the option to publish the peer review history of their article (what does this mean?). If published, this will include your full peer review and any attached files.

**Do you want your identity to be public for this peer review?** For information about this choice, including consent withdrawal, please see our Privacy Policy.

Reviewer #1: No

Reviewer #2: No
